# Phases and dynamics of quantum droplets in the crossover to two-dimensions

Jose Carlos Pelayo[1*], George Bougas[2], Thomás Fogarty[1], Thomas Busch[1] and Simeon I. Mistakidis[2,3]

**1** Quantum Systems Unit, Okinawa Institute of Science and Technology Graduate University, Okinawa, Japan 904-0495

**2** Department of Physics, Missouri University of Science and Technology, Rolla, MO 65409, USA

**3** ITAMP, Center for Astrophysics | Harvard & Smithsonian, Cambridge, USA

⋆ jose.pelayo@oist.jp

## ABSTRACT

We explore the ground states and dynamics of ultracold atomic droplets in the crossover region from three to two dimensions by solving the two-dimensional and the quasi two-dimensional extended Gross-Pitaevskii equations numerically and with a variational approach. By systematically comparing the droplet properties, we determine the validity regions of the pure two-dimensional description, and therefore the dominance of the logarithmic nonlinear coupling, as a function of the sign of the averaged mean-field interactions and the size of the transverse confinement. One of our main findings is that droplets can become substantially extended when their binding energies become small upon transitioning from negative-to-positive averaged mean-field interactions. To explore fundamental dynamical properties in the cross-over region, we study interaction quenches and show that the droplets perform a periodic breathing motion for modest quench strengths, while larger quench amplitudes lead to continuous expansion exhibiting density ring structures. We also showcase that it is possible to form complex bulk and surface density patterns in anisotropic geometries following the quench. Since we are working with realistic parameters, our results can directly facilitate future experimental realizations.

## CONTENTS

# I. INTRODUCTION

Bound state formation is a fundamental and common process in many different areas ranging from nuclei formation in high-energy physics [1, 2] via Cooper pair formation in condensed matter [3, 4], to, for example, Efimov state [5, 6] and droplet formation in ultracold atomic systems [7–9]. The latter are ultradilute states of many-body quantum matter with well-defined surface tension [10–12], which owe their existence to the balance between mean-field interactions and quantum fluctuations [7]. They have been realized in a series of recent cold atom experiments using either dipolar $^{164}$Dy [13, 14] or $^{166}$Er [15, 16] atoms, as well as mixtures thereof [17]. They have also been observed in mixtures with attractive short-range intercomponent interactions made from different $^{39}$K hyperfine states [18–20] or distinct atoms such as $^{87}$Rb and $^{41}$K [21, 22] or $^{87}$Rb and $^{23}$Na [23]. All these systems allow one to tune the relevant parameters into the droplet regimes [16, 24, 25], where the quantum corrections usually correspond to the perturbative Lee-Huang-Yang (LHY) contribution [26] (see also Refs. [27–30] regarding beyond LHY correlation aspects). The appropriate inclusion of the LHY term, being itself dimension dependent [31–33], into the dynamical equations leads to system-specific extended Gross-Pitaevskii (eGPE) frameworks [7, 31].

Over the past couple of years a plethora of droplet properties have been discussed based on the relevant eGPEs. These include their inelastic collisions [34–36], their collective excitations in both one- [37, 38] and two-dimensions [39, 40], their self-evaporation process which prevents them from sustaining collective modes [41], and their ability to coexist with solitons [42, 43], vortices [44–46], impurities [47–51] and dispersive shock-waves [52, 53]. It is interesting to note that droplets can occur in lattice systems [54, 55], in the presence of spin-orbit coupling [56, 57], but also beyond bosonic systems e.g. in Bose-Fermi mixtures [58, 59]. The widely employed parameter region in which droplets have been primarily studied and in which these systems are stable [19] is defined by a fixed intercomponent density ratio dictated by the underlying intracomponent couplings. In these regions the two-component bosonic system can be reduced to an effective single-component one (alias a symmetric droplet) [7, 31], but mass- and intracomponent interaction imbalanced droplet configurations have also been studied [60–63].

Importantly, the LHY term describing the quantum fluctuations has different forms in different spatial dimensions as well as in the crossover regions [32, 33, 64, 65]. In particular, it appears to be repulsive (attractive) in three-dimensions (one-dimension) and scales with the density as $\sim n^{5/2}$ ($\sim n^{3/2}$), requiring that the mean-field interactions are attractive (repulsive) to ensure an energy minimum corresponding to a droplet solution. It should be emphasized that the current understanding of the droplet physics in the regions of dimensional crossover and in general of the regions of validity of the respective eGPEs is far from complete and has only been partially discussed [32, 33, 64]. It is therefore valuable, especially for future experimental efforts, to determine the parameter regimes where different eGPE descriptions are valid, as well as confirming that in the crossovers different terms predict the same behavior. For this we consider in the following a symmetric bosonic droplet with contact interactions in the crossover from three dimensions (3D) to two dimensions (2D). The different interaction regimes we discuss can be experimentally assessed using Feshbach resonances in 3D [19] and we explore the ground state and the dynamical response of the droplet by solving the relevant eGPE in pure 2D and in quasi-2D geometries.

Starting from the underlying energy functionals, we first discuss the 2D and quasi-2D eGPEs and focus on the LHY contribution, the mean-field interactions and the chemical potential with respect to the 3D scattering lengths. We specifically consider finite sized systems with the intercomponent density ratios determined by the intracomponent interaction ratios, rather than equal particle numbers and interactions that have been primarily considered so far. In addition to the numerical solutions of the eGPEs, we also construct a variational ansatz for both geometries, which allows us to determine the experimentally relevant parameter regions of validity of the two regimes. We show that for the 2D description to hold for positive averaged mean-field interactions, a wide range of transversal confinement strength can be allowed for, while for negative averaged mean-field interactions tight transverse confinements are required. For positive mean-field interactions this can be traced back to the dominant role of the logarithmic LHY term. The validity regions are further verified by a direct comparison of different observables, such as energy per particle, the droplet density and width, by comparing the predictions of the respective eGPEs.

We also consider interaction quenches in the crossover regions and find that the droplets can undergo a collective breathing motion, whose frequency decreases for larger post-quench coupling strengths. In the quasi-2D regime, the breathing frequency and amplitude is smaller than in the 2D case, yet once the quench amplitude increases enough, the droplet solely expands and features ring-shaped excitation patterns. In general, the collective response after an interaction quench is a mixture of excitations of the monopole and quadrupole modes for relatively small quench amplitudes, whereas bulk and surface density patterns are triggered for increasing post-quench interactions and in

anisotropic settings.

Our manuscript is organised as follows. In Sec. II, we introduce the droplet system and explain the differences of the eGPEs in 2D and in quasi-2D. We also describe the variational analysis used to complement the direct numerical solutions of the eGPEs. Next, in Sec. III, we address the ground state properties of the droplet configurations in the dimensional crossover by directly comparing the predictions of the two eGPEs for different choices of the intercomponent scattering lengths and the size of the transversal confinement. This allows us to determine the limits of the applicability of the pure 2D description. Sec. IV elaborates on the dynamical response of droplets to an interaction quench by discussing collective motion and excitation patterns. Finally, we analyze the impact of trap anisotropy within the radial plane onto the ground state and the collective excitations in Sec. V. We summarize our findings and propose future extensions of our work in Sec. VI. For completeness, Appendix A provides among the full 3D eGPE and quasi-2D and 2D ones within the respective variational approaches.

## II.   QUASI TWO-DIMENSIONAL ATTRACTIVE BOSONIC MIXTURES

We consider a homonuclear ($m_1 = m_2 \equiv m$) bosonic mixture in the $|1\rangle \equiv |F = 1, m_F = -1\rangle$ and $|2\rangle \equiv |F = 1, m_F = 0\rangle$ hyperfine states of $^{39}$K as in the recent 3D experiments of Refs. [19, 20]. The atoms feature intracomponent repulsion characterized by the 3D $s$-wave scattering lengths $a_{11} \neq a_{22} > 0$ and intercomponent attraction, $a_{12} < 0$. By varying the relevant magnetic field slightly below 57G by means of a Feshbach resonance [66], one can reach an interaction regime where $\delta a = a_{12} + \sqrt{a_{11}a_{22}} \lesssim 0$, thus permitting droplet formation in the presence of quantum fluctuations which act repulsively and stabilize the system. Below we assume the experimentally verified condition $n_1/n_2 = \sqrt{a_{22}/a_{11}}$ [19], such that the droplet description reduces to a single-component one [7, 31, 67], with $n_i$ denoting the density of the $i$-th component.

We assume that our effectively single-component droplets are trapped in an extensive box potential of length $L_x = L_y \equiv L$ in the $x$-$y$ plane and a tight box of length $L_z$ in the $z$ direction. The hard walls of the box in the plane are placed far away such that they do not impact the droplet structures. In the following, we investigate the fundamental differences between the pure 2D and quasi-2D regimes by solving for the ground states the underlying eGPEs numerically and by using a variational ansatz. This will allow us to determine the regions of validity of the two descriptions.

### A.   Extended Gross-Pitaevskii models in 2D and in quasi-2D

To properly describe droplets in the 2D crossover we first present the corresponding eGPE frameworks in pure 2D and in quasi-2D. The parameter regimes we discuss are chosen to match the current state-of-the-art droplet experiments [19, 20], which is one of the aspects that places our findings beyond previous works [32, 33, 64]. There are two main facts in the dimensional crossover that need to be taken into account: i) in pure 2D, the atomic motion in the transverse direction is essentially frozen while in quasi-2D transverse excitations can take place and ii) the LHY correction is different in the different dimensions. While we show below that both of these facts are crucial to model the droplet, we also argue that parameter regions exist in which both descriptions are valid.

In pure 2D, where the transversal energy gap is much larger than the chemical potential, the energy functional of the effective single-component droplet takes the form [31, 63, 68–70]

$$E_{2D}[\psi] = \int d\mathbf{r} \left\{ \frac{\hbar^2}{2m} |\nabla\psi|^2 + \frac{\hbar^2}{2m} g|\psi|^4 \ln\left[\frac{|\psi|^2}{n_0^{(2D)}e}\right] \right\}. \tag{1}$$

Here, the 2D droplet wave function $\psi(\mathbf{r}) = \psi(x, y)$ relates to the individual components as $\psi(x, y) \equiv \psi_1\sqrt{(1 + \lambda)} = \psi_2\sqrt{(1 + 1/\lambda)}$, where the renormalization parameter $\lambda$ depends on the involved scattering lengths (see Eq. (4)).

The second term on the right hand side in the above expression accounts for the combined mean-field and LHY interaction contributions. This is characteristic for the 2D case [8], while in 3D [7] and 1D [30, 31] the interaction energies stemming from different physical origins are captured by separate terms. The effective interaction coefficient

can be written as

$$g = \frac{4\pi}{\ln\left[\dfrac{a_{12}^{(2D)}\sqrt{a_{11}^{(2D)}a_{22}^{(2D)}}}{\left[a_{11}^{(2D)}\right]^2 \Delta}\right]\ln\left[\dfrac{a_{12}^{(2D)}\sqrt{a_{11}^{(2D)}a_{22}^{(2D)}}}{\left[a_{22}^{(2D)}\right]^2 \Delta}\right]}, \tag{2a}$$

$$\Delta = \exp\left\{-\frac{\ln^2\left(\dfrac{a_{22}^{(2D)}}{a_{11}^{(2D)}}\right)}{2\ln\left(\dfrac{[a_{12}^{(2D)}]^2}{a_{11}^{(2D)}a_{22}^{(2D)}}\right)}\right\}, \tag{2b}$$

with $a_{ij}^{(2D)}$ referring to the respective 2D $s$-wave scattering lengths. The latter are related to their 3D counterparts (which are experimentally tunable) via the mapping [32]

$$a_{ij}^{(2D)} = 2L_z e^{-\gamma - L_z/(2a_{ij})}, \tag{3}$$

where $\gamma = 0.577\ldots$ is the Euler-Mascheroni constant [71]. This mapping enables one to express the parameter $\lambda$ in terms of the 3D scattering lengths as

$$\lambda = \sqrt{\frac{\ln\left(\dfrac{a_{12}^{(2D)}\sqrt{a_{11}^{(2D)}a_{22}^{(2D)}}}{\left[a_{22}^{(2D)}\right]^2 \Delta}\right)}{\ln\left(\dfrac{a_{12}^{(2D)}\sqrt{a_{11}^{(2D)}a_{22}^{(2D)}}}{\left[a_{11}^{(2D)}\right]^2 \Delta}\right)}} = \sqrt{\frac{a_{11}}{a_{22}}\left(\frac{a_{12}-a_{22}}{a_{12}-a_{11}}\right)}. \tag{4}$$

Moreover, $n_0^{(2D)}$ is the droplet saturation density [8, 31] in the thermodynamic limit given by

$$n_0^{(2D)} = \frac{e^{-2\gamma-3/2}}{\pi a_{12}^{(2D)}\sqrt{a_{11}^{(2D)}a_{22}^{(2D)}}}\Delta\sqrt{\frac{4\pi}{g}}. \tag{5}$$

Demanding that the energy functional is stationary upon small variations of the wave function $\psi$ [72, 73], yields the corresponding 2D eGPE

$$i\hbar\frac{\partial\psi}{\partial t} = -\frac{\hbar^2}{2m}\nabla^2\psi + \frac{\hbar^2}{m}g|\psi|^2\psi\ln\left(\frac{|\psi|^2}{n_0^{(2D)}\sqrt{e}}\right), \tag{6}$$

where the droplet wave function $\psi$ is normalised to the total particle number, i.e. $\int d\mathbf{r}\,|\psi(\mathbf{r})|^2 = N = N_1 + N_2$. For large $gN$ [68], this eGPE describes finite droplets with a flat-top density core $n$ which drops to zero at the edges over a characteristic length set by the healing length $\xi$. The latter can be estimated from the chemical potential, $\mu_{2D}$, as $\xi \sim \hbar/\sqrt{m|\mu_{2D}|} = 1/\sqrt{gn\left|\ln\left(n/(n_0^{(2D)}\sqrt{e})\right)\right|}$ [31]. For adequately large particle numbers the thermodynamic limit is reached, and a homogeneous profile $n \simeq n_0^{(2D)}$ is obtained.

Turning to the corresponding quasi-2D setup, where a tight transversal box potential of length $L_z$ with ground state energy $\epsilon_0 = 2\pi^2\hbar^2/mL_z^2$ is present, the LHY term is given by a complicated integral expression, resulting in an integro-differential eGPE [32]. The latter depends on the dimensionless ratio of the intracomponent mean-field interaction energy of each of the two components with respect to the box energy in the transverse direction

$$\chi = \frac{4\pi\hbar^2}{m}\left(\frac{a_{11}|\Psi_1|^2 + a_{22}|\Psi_2|^2}{\epsilon_0}\right) = \frac{4\pi\hbar^2}{m}\sqrt{a_{11}a_{22}}\frac{|\Psi|^2}{\epsilon_0}. \tag{7}$$

To derive the expression on the right hand side, we have assumed a fixed density ratio $n_1/n_2 = \sqrt{a_{22}/a_{11}}$ with which the effective single-component wave functions take the form $\Psi(x,y,z) = \sqrt{(1+\sqrt{a_{11}/a_{22}})}\Psi_1(x,y,z) = \sqrt{(1+\sqrt{a_{22}/a_{11}})}\Psi_2(x,y,z)$. When the excitation energies in the transversal direction are much larger than the

in-plane (radial) excitation energies, which corresponds to $\chi \ll 1$, single particle excitations take place in the plane while along the transverse direction predominantly collective motion occurs. Within this limit, an approximate analytic LHY contribution can be found having the form [32]

$$E_{LHY}^{(q2D)}[\chi] = \int d\mathbf{r} dz \frac{\pi}{4} \chi^2 \left[ \ln\left(2\pi^2 \chi \sqrt{e}\right) + \frac{\pi^2}{3} \chi \right]. \tag{8}$$

It can be shown that Eq. (8) matches well with the aforementioned integral as long as $\chi \lesssim 0.3$. Using the expression in Eq. (7), the full energy functional is then given by

$$E_{q2D}[\Psi] = \int dz d\mathbf{r} \left\{ \frac{\hbar^2}{2m} |\nabla \Psi|^2 + |\Psi|^4 \frac{4\pi \hbar^2 \delta a}{m} \frac{\sqrt{a_{22}/a_{11}}}{(1 + \sqrt{a_{22}/a_{11}})^2} \right. \tag{9}$$
$$\left. + \underbrace{\frac{2\pi \hbar^2 a_{11} a_{22}}{m L_z} |\Psi|^4 \ln\left(4\pi |\Psi|^2 \sqrt{a_{11} a_{22} e} L_z^2\right)}_{E_{LHY}^{(1)}} + \underbrace{\frac{4\pi^2 \hbar^2}{3m} L_z |\Psi|^6 (a_{11} a_{22})^{3/2}}_{E_{LHY}^{(2)}} \right\}.$$

Here the second term represents the combination of all mean-field energy contributions, whereas the third, $E_{LHY}^{(1)}$, and fourth, $E_{LHY}^{(2)}$, terms stem from the explicit LHY energy of Eq. (8). Specifically, $E_{LHY}^{(1)}$ exhibits a logarithmic non-linearity similar to the 2D setup, while $E_{LHY}^{(2)}$ is a higher-order term absent in the 2D setup. Notice that the first LHY contribution is negative for $\delta a \gtrsim 0$ (see also the discussion in Sec. III) which is a specific feature of the quasi-2D setting when compared to the 3D one [32].

Applying a variational approach to the above energy functional results in the quasi-2D eGPE

$$i\hbar \frac{\partial \Psi}{\partial t} = -\frac{\hbar^2}{2m} \nabla^2 \Psi + \frac{8\pi \hbar^2 \delta a}{m} \frac{\sqrt{a_{22}/a_{11}}}{(1 + \sqrt{a_{22}/a_{11}})^2} |\Psi|^2 \Psi \tag{10}$$
$$+ \frac{4\pi \hbar^2 a_{11} a_{22}}{m L_z} |\Psi|^2 \Psi \ln\left(4\pi e |\Psi|^2 \sqrt{a_{11} a_{22}} L_z^2\right) + \frac{4\pi^2 \hbar^2}{m} L_z |\Psi|^4 \Psi (a_{11} a_{22})^{3/2},$$

where again the second term represents the combination of intra- and intercomponent mean-field energies, and the third and fourth terms stem from the LHY correction. Notice that, even though there is a logarithmic dependence (third term) similar to the pure 2D setup, a higher-order non-linearity ($\sim |\Psi|^4 \Psi$) appears as well. This additional nonlinearity modifies the behavior of the characteristic (healing) length over which the constant flat-top density falls to zero, i.e. $\xi \sim \hbar/\sqrt{m|\mu_{q2D}|}$, where the chemical potential for a constant density $n$ is now given by

$$\mu_{q2D} = \frac{8\pi \hbar^2 \delta a}{m} \frac{\sqrt{a_{22}/a_{11}}}{(1 + \sqrt{a_{22}/a_{11}})^2} n + \frac{4\pi \hbar^2 a_{11} a_{22}}{m L_z} n \ln\left(4\pi e n \sqrt{a_{11} a_{22}} L_z^2\right) \tag{11}$$
$$+ \frac{4\pi^2 \hbar^2}{m} L_z n^2 (a_{11} a_{22})^{3/2}.$$

As it can be seen, the chemical potential in quasi-2D and 2D scales with the density as $\sim n \ln(n)$. However, in quasi-2D two additional terms appear. The first one with a linear density dependence originates from the mean-field contribution, whereas the second is a higher-order density term ($\sim n^2$) stemming from the LHY modification in quasi-2D. It is immediately clear that the latter term will become suppressed for spatially extended droplet structures in the $x-y$ plane (see Sec. III). Through the zero-pressure requirement ($E_{q2D} - \mu_{q2D} n = 0$) [31, 32, 58], one obtains the equilibrium density in the thermodynamic limit as

$$n_0^{(q2D)} = \frac{3}{4\pi \sqrt{a_{11} a_{22}} L_z^2} \mathcal{W}\left( \frac{e^{-3/2}}{3} \exp\left( \frac{-2\delta a L_z}{a_{11} a_{22}} \frac{\sqrt{a_{22}/a_{11}}}{(1 + \sqrt{a_{22}/a_{11}})^2} \right) \right), \tag{12}$$

where $\mathcal{W}(z)$ is the Lambert W-function [71] [1]. In the case where the higher order term ($\sim n^2$) in the quasi-2D eGPE can be neglected, that is, for $\delta a > 0$ as will be shown in Fig. 3(b), one retrieves the equilibrium density calculated in Ref. [32].

—————

[1] Notice that expressing the energy functional in terms of the equilibrium density results in the implicit form

$$E_{q2D}[\Psi] = \int dz d\mathbf{r} \left\{ \frac{\hbar^2 |\nabla \Psi|^2}{2m} + \frac{2\pi \hbar^2 a_{11} a_{22}}{m L_z} |\Psi|^4 \ln\left( \frac{|\Psi|^2}{n_0^{(q2D)} e} \right) + \frac{4\pi^2 \hbar^2}{3m} (a_{11} a_{22})^{3/2} L_z |\Psi|^4 (|\Psi|^2 - 2n_0^{(q2D)}) \right\}.$$

Let us note that we use the imaginary time propagation method for the numerical calculation of the ground states of the eGPEs and that a spatial (temporal) discretization $dx = dy = 0.1$ ($dt = 10^{-3}$) ensures that the relative energy deviation during any time evolution remains of the order of $10^{-12}\,\%$ ($10^{-9}\,\%$) in 2D (quasi-2D).

## B. Variational approach

In addition to directly comparing the results obtained by numerically solving the eGPEs in 2D and quasi-2D, we will also use a variational method to gain further insight on the droplet properties in the dimensional crossover region. Such an approach has been previously shown to quantitatively describe the droplet-to-soliton transition in a quasi-2D setting when varying the confinement length of the box [65]. We assume that due to the strong transverse confinement, the wave function $\Psi(x,y,z)$ can be written as a product of a radial profile $\psi(x,y)$ and a transverse one, $\varphi(z) \simeq L_z^{-1/2}$, i.e. $\Psi(x,y,z) = \psi(x,y)\varphi(z)$. Such an ansatz is motivated by the actual numerically obtained ground state density along the $z$-direction from Eq. (10), which displays a homogeneous profile.

For the radial part of the wave function we choose a super-Gaussian trial wave function that depends on two variational parameters [68, 70, 74–76]

$$\psi(x,y) = \frac{\sqrt{N}}{\sigma_r \sqrt{\pi \Gamma(1 + 1/m_r)}} e^{-\frac{1}{2}\left(\sqrt{x^2+y^2}/\sigma_r\right)^{2m_r}},\tag{13}$$

where $\sigma_r$ characterizes the spatial extent of the droplet and the exponent $m_r > 0$ accounts for its "flatness". Evidently, for $m_r = 1$ a Gaussian distribution is retrieved, while the limit of $m_r \to \infty$ describes a constant homogeneous density. Inserting this ansatz into the energy functional of the quasi-2D system (Eq. (9)) leads to the following expression for the energy per particle in the droplet

$$\begin{aligned}
\frac{E_{q2D}}{N} =\ & \frac{\hbar^2}{2m} \frac{m_r^2}{\sigma_r^2 \Gamma(1/m_r)} + \frac{N 2^{-1/m_r}}{\pi \sigma_r^2 \Gamma(1 + 1/m_r)} \frac{2\pi\hbar^2}{mL_z} \left[ \frac{\sqrt{a_{22}/a_{11}}}{(1 + \sqrt{a_{22}/a_{11}})^2} 2\delta a \right. \\
& \left. + \frac{a_{11}a_{22}}{L_z} \ln\left(\frac{N 4\pi \sqrt{a_{11}a_{22}e}L_z}{\sigma_r^2 \pi \Gamma(1 + 1/m_r)}\right) \right] + \frac{4\pi^2 \hbar^2}{3mL_z} \frac{N^2 3^{-1/m_r}}{\pi^2 \sigma_r^4 \Gamma^2(1 + 1/m_r)} (a_{11}a_{22})^{3/2} \\
& - N \frac{2\pi\hbar^2 a_{11}a_{22}}{mL_z^2} \frac{2^{-1-1/m_r}}{\pi \sigma_r^2 \Gamma(1/m_r)}.
\end{aligned}\tag{14}$$

From this the variational parameters $\sigma_r$ and $m_r$ can be determined by minimization under the constraint of the given external parameters $a_{ij}$, $N$, and $L_z$. This allows one to express the dimensionless $\chi_{q2D}$ parameter in the simple form

$$\chi_{q2D} = \frac{2N}{\sigma_r^2 \pi^2 \Gamma(1 + 1/m_r)} \sqrt{a_{11}a_{22}} L_z.\tag{15}$$

Carrying out the same procedure in the pure 2D setting, we find that $\chi_{2D}$ has the same functional form as $\chi_{q2D}$, however, the parametrization of the variational parameters is different, as they stem from the minimization of a different energy per particle given by (see also Eq. (1))

$$\frac{E_{2D}}{N} = \frac{\hbar^2}{2m} \frac{m_r^2}{\sigma_r^2 \Gamma(1/m_r)} - \frac{\hbar^2}{2m} \frac{gN 2^{-1/m_r}(m_r + 1)}{2\pi \sigma_r^2 \Gamma(1/m_r)} + \frac{\hbar^2}{2m} \frac{gN 2^{-1/m_r} m_r}{\pi \sigma_r^2 \Gamma(1/m_r)} \ln\left(\frac{N m_r}{\pi \sigma_r^2 \Gamma(1/m_r)\sqrt{e}n_0^{(2D)}}\right).\tag{16}$$

Comparing the energies in 2D given by Eq. (14) and in quasi-2D described by Eq. (16), it becomes evident that they both scale with $\sigma_r^{-2}$ and $\sigma_r^{-2}\ln(\sigma_r^{-2})$. However, in quasi-2D there is an additional higher-order density term depending on $N^2\sigma_r^{-4}$, which only becomes negligible for droplets with large radial extent.

## III. GROUND STATE PROPERTIES IN THE 2D AND QUASI-2D REGIMES

In the following, we will first identify the parameter regions of $\delta a$ and $L_z$ where the analytic expression of the quasi-2D energy functional applies. As mentioned above, the expression in Eq. (9) holds for $\chi_{q2D} \lesssim 0.3$, which also includes the crossover to the pure 2D regime reached for $\chi_{q2D} \ll 1$. To achieve a clean comparison with the 2D geometry we use for both cases the underlying 3D scattering lengths which translate into their 2D counterparts as

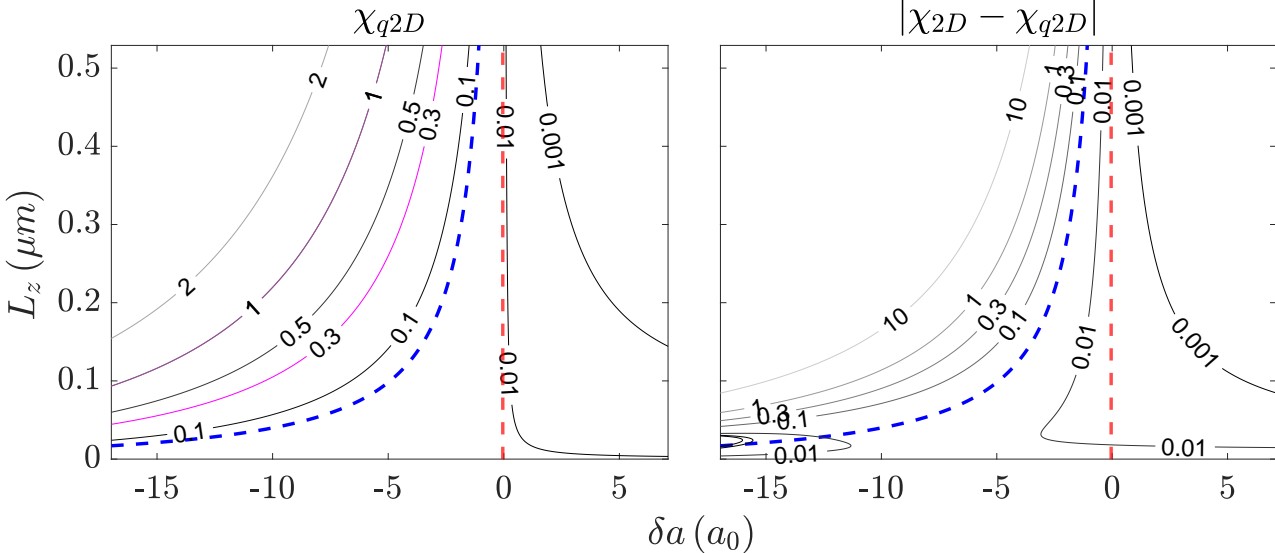

FIG. 1. (a) Ratio of the averaged mean-field intracomponent interaction and the transverse box energies, measured through $\chi_{q2D}$ (Eq. (15)) and being obtained via the quasi-2D variational approach. The area where $\chi_{q2D} < 0.05$ (below the blue dashed line) indicates the parameter regions where the quasi-2D setting can be regarded as 2D. (b) This agreement is further quantified by the absolute difference $|\chi_{q2D} - \chi_{2D}|$ evaluated using the corresponding variational models. Also here a good match between the quasi-2D and 2D models occurs below the blue dashed curve, extracted by comparing the equilibrium droplet densities in the thermodynamic limit (see also text). To guide the eye we have marked the $\delta a = 0$ threshold by a red dashed line. In all cases $N = 2 \times 10^5$.

given in Eq. (3) for different transverse box lengths $L_z$. In Fig. 1(a) we show $\chi_{q2D}$ obtained with the variational method, where the region of validity of the quasi-2D description lies below the magenta contour line corresponding to $\chi_{q2D} = 0.3$. One can immediately note that for $\delta a > 0$ (on the right side of the red dashed line in Fig. 1(a)) the quasi-2D energy functional described by Eq. (9) holds even for large transverse box sizes. In contrast, for $\delta a < 0$ the quasi-2D LHY term remains valid only for tight transverse box confinement, a behavior that is more pronounced for stronger attractive $\delta a$ values.

Analytical insight regarding the crossover from quasi-2D to 2D can be obtained in the thermodynamic limit by demanding that the normalized difference in equilibrium densities is small, namely $\left( \left| n_0^{(2D)} - n_0^{(q2D)} \right| \right) / n_0^{(2D)} \leq 0.2$ [32]. This results in $L_z = -4a_{11}a_{22}/\delta a$, depicted as the blue dashed line in Fig. 1(a). Below that line $\chi_{q2D}$ becomes very small, i.e. $\chi_{q2D} \lesssim 0.05$, and therefore the quasi-2D eGPE description becomes equivalent to the pure 2D one. This can be further confirmed by looking at the difference $|\chi_{q2D} - \chi_{2D}|$, as predicted from the variational quasi-2D and 2D calculations in Fig. 1(b), which also gives small values in the region separated by the blue dashed curve, indicating that the equilibrium density estimate is adequate. Let us note that a comparison with the 3D eGPE in the regions where $\chi_{q2D} \gtrsim 0.3$ is an interesting perspective for future investigations and we briefly discuss the ground state energies obtained from the variational approach of the 3D, quasi-2D and 2D eGPEs in Appendix A.

To gain a better understanding of the differences between the 2D and the quasi-2D descriptions, we next carefully inspect the distinct energy terms in the respective energy functionals given by Eqs. (1) and (9). In the case of attractive averaged mean-field interactions, i.e. $\delta a < 0$, the mean-field contribution in the quasi-2D case (second term in Eq. (9)) dominates over the logarithmic and higher-order nonlinear energy terms (third and fourth contributions in Eq. (9) respectively), see Fig. 2(a) and its inset. This naturally results in a large magnitude of $\chi_{q2D}$, and hence is consistent with the deviations from the 2D model, where the logarithmic LHY term is always present. On the other hand, for positive $\delta a$ the logarithmic nonlinearity is the leading-order contribution to the quasi-2D energy functional, see Fig. 2(b) and its inset. In this parameter regime the quasi-2D and the 2D approaches are therefore practically described by the same LHY contribution, which is precisely the parameter regime where $|\chi_{q2D} - \chi_{2D}| \simeq 0$ as can be clearly seen in Fig. 1(b). Finally, one can see from Fig. 2 that regardless of the sign of $\delta a$, the higher-order nonlinear term $\sim (a_{11}a_{22})^{3/2}$ in the quasi-2D energy functional is suppressed, and that for $\delta a < 0$ the mean-field (logarithmic LHY) interaction energy is negative (positive), while for $\delta a > 0$ this is reversed.

Focusing on the interaction regimes where agreement between the quasi-2D and the 2D approach can be expected from the variational considerations, we next investigate the binding energies and spatial widths of the droplets. The

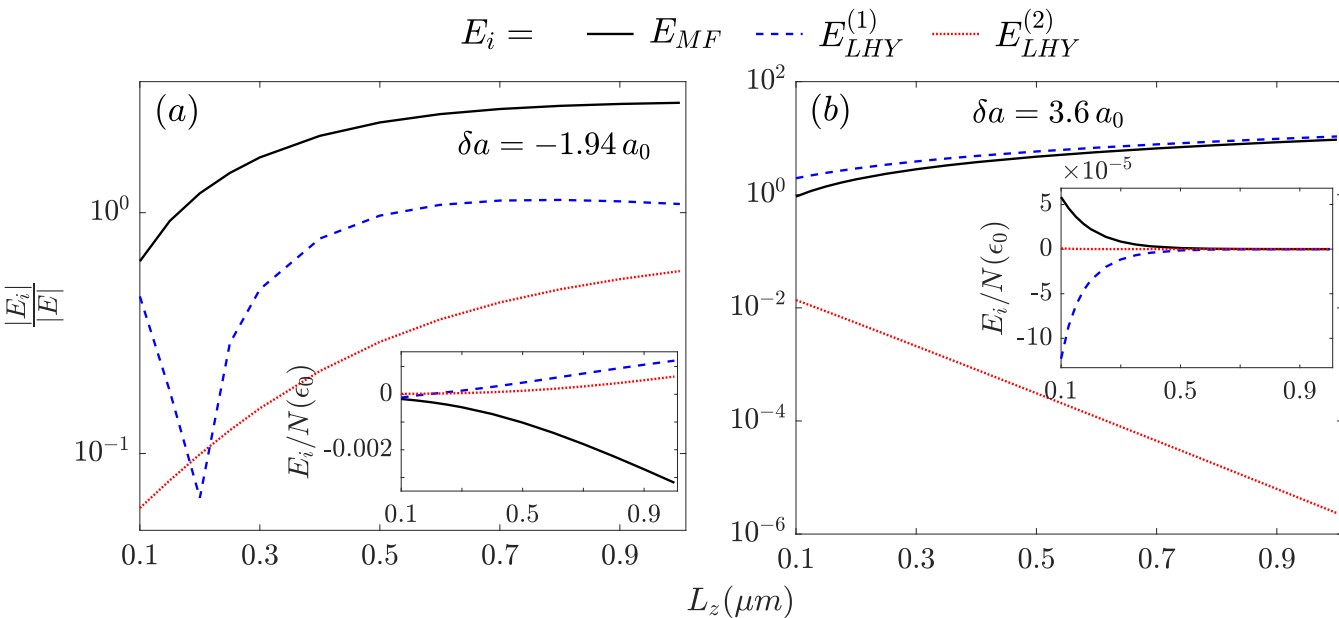

FIG. 2. Magnitudes of the different energy contributions, $E_i$, scaled by the total energy, $E$, of the droplet within the quasi-2D eGPE model for (a) attractive $\delta a < 0$ and (b) repulsive $\delta a > 0$, as a function of the transverse box length $L_z$. One can see that in the case of $\delta a < 0$ ($\delta a > 0$) the leading energy term is $E_{MF}$ ($E_{LHY}^{(1)}$), and that, independently of the sign of $\delta a$, the higher-order quantum fluctuation term $E_{LHY}^{(2)}$ possesses the smallest magnitude. The insets depict the individual energy terms per particle, i.e. $E_i/N$, from which the change of sign of the LHY term when crossing $\delta a = 0$ can be seen. Note that the initial dip in $|E_{LHY}^{(1)}|$, which is visible in panel (a), is due to $E_{LHY}^{(1)}$ being negative for $L_z \lesssim 0.2\mu m$ and positive otherwise (see inset). All other system parameters are as in Fig. 1.

energy per particle, $E/N$, as obtained from the variational quasi-2D and 2D approaches but also from the numerical solution of the quasi-2D eGPE is shown in Fig. 3(a) as a function of $L_z$ and for specific $\delta a$ interactions. Remarkably, for the quasi-2D eGPE the variational and the numerical approach give quantitatively the same results, regardless of $\delta a$ and $L_z$. Moreover, the energy per particle remains negative for either attractive or repulsive averaged mean-field interactions $\delta a$ since droplets are sustained independently of $|\delta a|/\sqrt{a_{11}a_{22}} < 1$ in a quasi-2D geometry [31, 32]. This is in contrast to the behaviour in 3D, where droplets can only form for $\delta a < 0$ [7]. However, quasi-2D and 2D droplets become less tightly bound for increasing $\delta a > 0$, especially for large $L_z$, where $|E|/N$ decreases (see Fig. 3(a)). This behavior can be attributed to the mean-field and logarithmic LHY energy terms, i.e. the second and third terms in Eq. (9), effectively cancelling each other in this interaction regime, which is shown in the inset of Fig. 2(b).

Additionally, rather good agreement is observed regarding the droplets binding energies for $\delta a > 0$ between the variational quasi-2D and 2D predictions, see solid and dashed lines in Fig. 3(a). This is a consequence of the dominant logarithmic LHY correction which is consistent with the relatively small values of $\chi_{q2D}$ in this regime depicted in Fig. 1(a). In contrast, for negative $\delta a$ (see $\delta a = -1.94\,a_0$ in Fig. 3(a)) more prominent deviations occur between the two approaches since in this interaction regime the mean-field energy (second term in Eq. (9)) dominates in quasi-2D while being absent in 2D. This disagreement is especially pronounced e.g. for $\delta a = -1.94\,a_0$ and $L_z \gtrsim 0.4\,\mu m$, where $\chi_{q2D} \sim 0.1$ and thus the quasi-2D model deviates from the pure 2D setup. This again confirms the threshold given by the blue-dashed line in Fig. 1(a) for any value of $\delta a$ and $L_z$. While we do not show this here, we have also confirmed that, similarly to the quasi-2D case, the predictions of the droplets binding energies from the 2D variational approach are in quantitative agreement with the ones of the numerical solution of the 2D eGPE (6).

Another important measure to characterize droplet configurations is their radial widths [70]

$$\sigma_r = \sqrt{\frac{2}{N} \int dx dy dz \, (x^2 + y^2)|\Psi(x, y, z)|^2}, \tag{17}$$

which is presented as a function of $L_z$ for different interaction strengths in Fig. 3(b). It can be readily deduced that the predictions of the quasi-2D variational approach are in accordance with the numerical results of the quasi-2D eGPE, and only start deviating from $L_z \gtrsim 0.6\,\mu m$ where the maximum relative deviations are of the order of 10%. More concretely, for $\delta a < 0$ the droplets have a small width due to the strongly bound character of the system, which

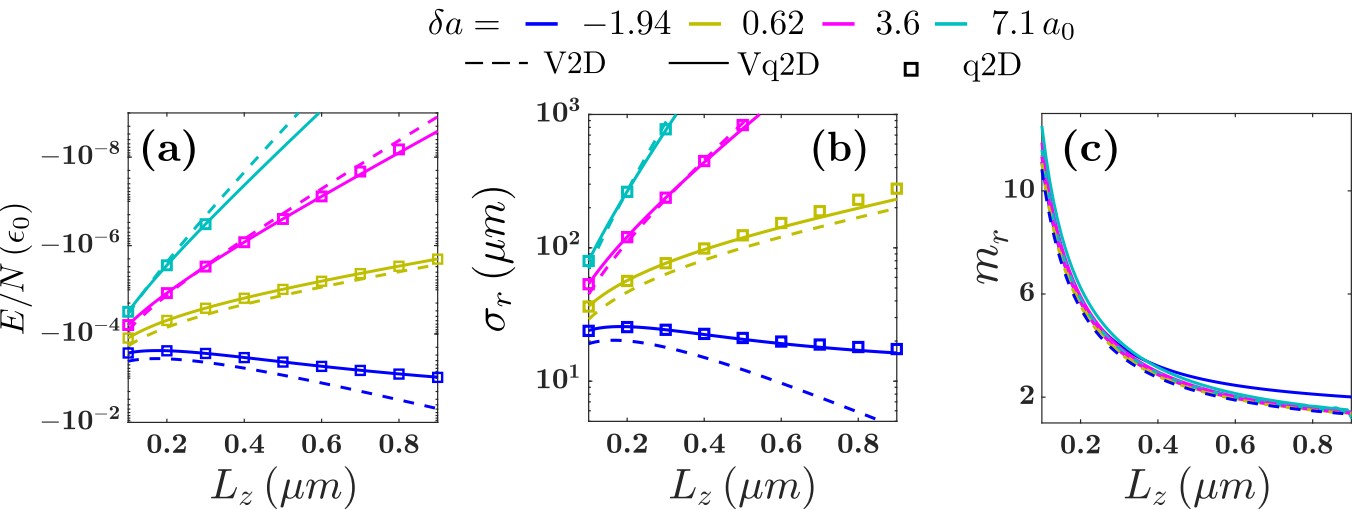

FIG. 3. (a) Energy per particle, $E/N$, and (b) radial width, $\sigma_r$, of the droplet obtained from the variational approaches in 2D (V2D) and quasi-2D (Vq2D) as well as from the numerical solution of the quasi-2D eGPE with respect to the transverse box length $L_z$ for different values of $\delta a$. Good agreement in both $E/N$ and $\sigma_r$ between the 2D and quasi-2D variational models can be seen, especially for $\delta a > 0$ and $L_z < 0.4\mu m$. Also, the predictions of the variational outcome and the numerical one in quasi-2D are in accordance. (c) The variational exponent $m_r$ as a function of $L_z$. It becomes apparent that the droplet's flat-top distribution becomes less pronounced for larger $L_z$ since $m_r$ decreases towards unity. The remaining system parameters are the same as in Fig. 1.

is also consistent with the large values of $|E|/N$ visible in Fig. 3(a). On the other hand, for repulsive $\delta a > 0$, these structures become more spatially extended across the $x$-$y$ plane, especially for large $L_z$, where small modifications in the transverse box confinement yield large increases in $\sigma_r$. This can be linked to the small negative energies per particle shown in Fig. 3(a), which means that these droplet configurations become less tightly bound and hence allow for large $\sigma_r$. Moreover, a comparison with $\sigma_r$ obtained within the 2D setting [Fig. 3(b)] reveals good quantitative agreement for $\delta a > 0$, which again can be attributed to the dominance of the logarithmic LHY correction in the repulsive interaction regime. In general, the droplets in a pure 2D configuration appear to be more tightly bound compared to their quasi-2D counterparts for $L_z < 0.2\,\mu m$, which is consistent with their slightly larger values of $|E|/N$ visible in Fig. 3(a).

To infer if the droplet density profile is Gaussian or flat-top shaped [8], we show the variational $m_r$ parameter in both the quasi-2D and the 2D settings in Fig. 3(c). As it can be seen, the droplet density distribution is strongly dependent on the transverse box length, but depends only weakly on the interactions $\delta a$. In particular, we observe that for small $L_z$ a larger $m_r$ indicates a flat-top density, whereas an increasing $L_z$ leads to a more Gaussian form.

The flat-top droplet profiles can also be confirmed by looking at the densities directly. For this we show cuts along $y = 0$ of the 2D densities $n_{2D}(x) = |\psi(x,0)|^2$ and of the transversally integrated quasi-2D densities $n_{q2D}(x) = \int dz|\Psi(x,0,z)|^2$ obtained from the numerical solutions of the respective eGPE in Fig. 4(a) for $L_z = 0.3\,\mu m$. These distributions do not only confirm the flat-top profiles, but also the monotonically increasing widths for increasing $\delta a$ as predicted by the variational calculations (see Fig. 3(b)-(c)). At the same time the peak density decreases, and better agreement between the quasi-2D and 2D can be observed for increasing $\delta a$. The latter is again consistent with the smaller deviations of the radial width of the droplet $\sigma_r$ between the two geometries as $\delta a$ turns positive (see Fig. 3(b)). It is worth mentioning that the flat-top shape is retained in both models upon $L_z$ variations as depicted in Fig. 4(b) for $\delta a = 3.6\,a_0$. Interestingly, a small increase in the transverse box length $L_z$ results in a substantially extended droplet size in the plane (see also Fig. 3(b)).

An additional crucial factor influencing the form of the density distribution of the ground state droplet distribution is the atom number $N$ [8, 18, 27, 65]. Using the variational approach, one can see from Fig. 5(a) and (b) that for attractive and repulsive mean-field regimes the negative energy per particle versus $L_z$ tends to an interaction-dependent constant value as $N$ increases. This behavior implies that the system reaches its thermodynamic limit, corresponding to $E/N = \mu_{2D} = -\hbar^2 g n_0^{(2D)}/(2m)$ in 2D [31, 68], where $g$ is given by Eq. (2a) and $E/N = \mu_{q2D}$ in quasi-2D (dotted lines in both panels in Fig. 5). These two values explain the discrepancy in the binding energy in the thermodynamic limit between 2D and quasi-2D. Notice here the progressively better agreement between the 2D and quasi-2D predictions for $\delta a > 0$ which is in line with our previous observations based on the eGPEs and different $N$ (Fig. 3(a)). However, for relatively small particle numbers such as $N \leq 2 \times 10^3$ the behavior of $E/N$ departs from

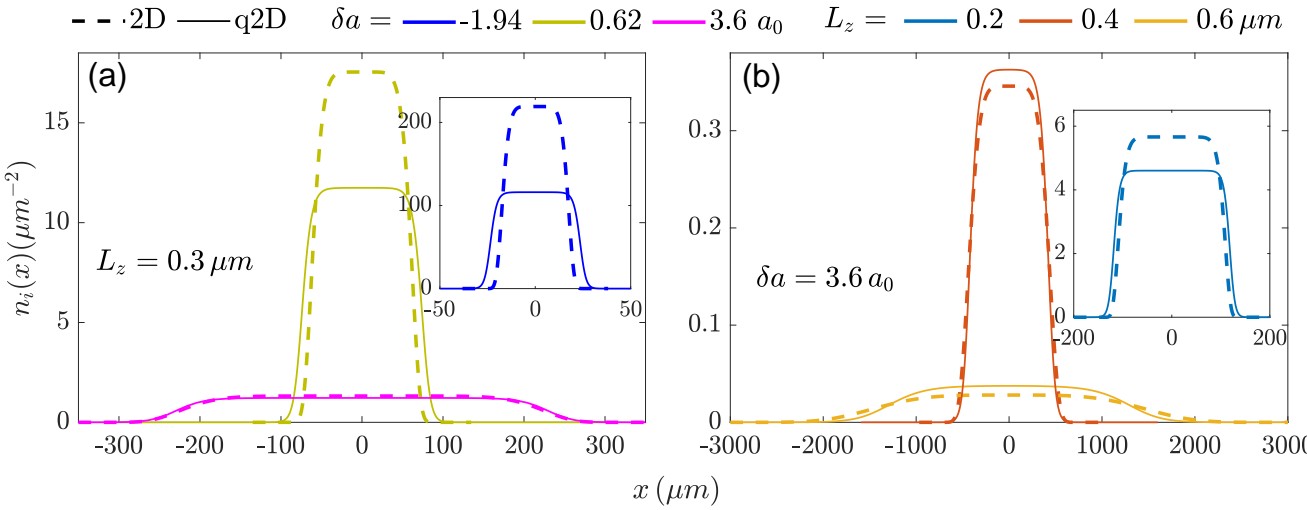

FIG. 4. (a) Ground state density profiles $n_i(x)$ of the droplet obtained from the $i \equiv$ 2D (at $y = 0$) and $i \equiv$ quasi-2D (at $y = 0$, integrated along $z-$ axis) eGPE models for fixed $L_z = 0.3\mu m$ and varying $\delta a$. It can be seen from panel (a) and its inset that for increasing $\delta a$ the droplets gradually increase their spatial extend. (b) The same as (a) but for constant $\delta a = 3.6a_0$ and different $L_z$. The agreement between the quasi-2D and 2D densities improves for either increasing $\delta a$ or larger $L_z$. The remaining system parameters are as in Fig. 1.

the thermodynamic limit as $L_z$ increases in both 2D and quasi-2D. In fact, in this case $E/N$ rapidly approaches very small values resulting in convergence issues in the variational calculations.

The radial width of the droplets for different particle numbers is depicted in Fig. 5(c), and (d). Evidently, it does not saturate but continuously grows with $N$, consistent with the incompressible character [7] of the droplet having a fixed flat-top value in the thermodynamic limit [8]. In fact, the rescaled quantity $\sigma_r/\sqrt{N}$ saturates for large $N$, similarly to $E/N$. Furthermore, it is apparent that for $\delta a < 0$ ($\delta a > 0$) $\sigma_r$ decreases (increases) for $L_z > 0.2\,\mu m$, reflecting the overall growth (reduction) of the droplet's binding quantified by $|E|/N$ as seen in Fig. 3(a). In accordance with the findings illustrated in Fig. 3(b), Fig. 5(d) shows that the variational results in 2D and quasi-2D align for $\delta a > 0$ up to $L_z \lesssim 0.4\,\mu m$, regardless of $N$. In contrast, for $\delta a < 0$, the radial width in the pure 2D case (dashed lines in Fig. 5(c), (d)) remains smaller than the quasi-2D outcome, irrespective of $N$. This can be attributed to the large droplet binding energy $|E|/N$ in 2D, see also Fig. 3(a).

## IV. QUENCH DYNAMICS IN THE 2D AND QUASI-2D REGIMES

Droplets in the 2D regime can support different collective modes both in the presence of an external trap [39] and in free space [68], but also nonlinear structures [44]. An intriguing question is therefore what the effect of transverse excitations and/or the distinct LHY contributions in the dimensional crossover is. We address this issue below by exploring the droplet dynamics after a quench of the averaged mean-field interactions from an initial value $\delta a_i$ to a final value $\delta a_f$ [27]. Specifically, we choose the initial droplet state to have $\delta a_i = 0.62\,a_0$ and $L_z = 0.2\,\mu m$, where according to our ground state analysis fairly good agreement occurs among the 2D and quasi-2D models (see Fig. 3(a) and (b)). In the following, we will show that the dynamics following a quench to $\delta a_f$ crucially depends on the quench amplitude $\delta a_f - \delta a_i$, and that for relatively small quench amplitudes, irrespective of the sign, it mostly corresponds to a collective breathing motion, while expansion occurs for large positive amplitudes.

Focusing first on small quench amplitudes, one can see from Fig. 6(a) that a quench of the form $\delta a_f > \delta a_i$ ($\delta a_f < \delta a_i$), triggers an initial expansion (contraction) of the quasi-2D droplet. This is consistent with the larger (smaller) radial width of the ground state droplet for the postquench interaction shown in Fig. 3(b). As time evolves, a collective breathing motion develops as evidenced by the oscillatory behavior of $\sigma_r(t)/\sigma_r(0)$, with the amplitude increasing for larger $|\delta a_f - \delta a_i|$. This response reflects the increasing energy difference $E(\delta a_i)/N - E(\delta a_f)/N$ (see Eq. (14)) for larger quench amplitudes, leading to significant variations in the kinetic energy of droplets, and hence in their width. The droplet breathing motion can also be seen in the expansion and contraction of the integrated

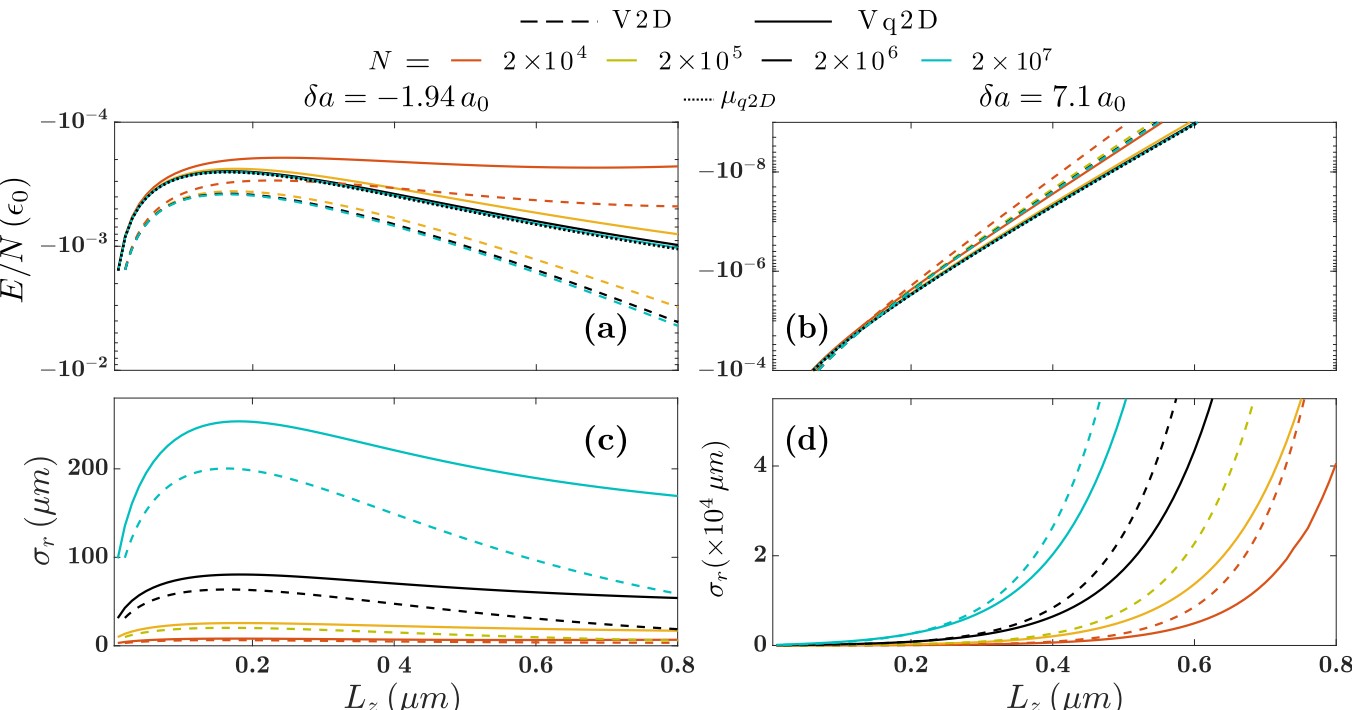

FIG. 5. Energy per particle, $E/N$, of the droplet stemming from the variational 2D and quasi-2D calculations for different total atom numbers at (a) $\delta a = -1.94a_0$ and (b) $\delta a = 7.1a_0$ as a function of the transverse box length. In the large particle limit, the quasi-2D $E/N$ approaches $\mu_{q2D}$ (dotted lines). (c), (d) Comparison of the droplet's radial width, $\sigma_r$, as predicted by the 2D and quasi-2D variational models for distinct $N$ for (c) $\delta a = -1.94a_0$ and (d) $\delta a = 7.1a_0$ as a function of $L_z$. It becomes evident that within the $\delta a > 0$ region, both $E/N$ and $\sigma_r$ show a similar qualitative behavior among the 2D and quasi-2D approaches.

density profiles $n_{q2D}(x,y) = \int dz |\Psi(x,y,z)|^2$ with respect to the initial droplet edge denoted by the white dashed lines in panels (a1)-(a5) of Fig. 7. Moreover, the oscillation amplitude of $\sigma_r(t)/\sigma_r(0)$ features a decreasing tendency at longer evolution times especially for $\delta a_f < \delta a_i$, see e.g. $\delta a_f = 0.49 a_0$ in Fig. 6(a). This indicates a comparatively weaker excitation of an additional mode, which in this case is given by a bulk mode within the droplet. Indeed, bulk excitations can be seen in $n_{q2D}(x,y)$ manifested as background density modulations, see e.g. the central depression with amplitude half of the peak density in Fig. 7(a3) or the ring structures in Fig. 7(a2) and (a4).

The decaying amplitude feature of $\sigma_r(t)/\sigma_r(0)$ becomes even more pronounced within the 2D eGPE model (dashed lines in Fig. 6(a)) as the bulk modes in the droplet density are enhanced compared to the quasi-2D eGPE (not shown). More importantly, however, appreciable deviations appear in the breathing frequency $\omega_b$, which is noticeably larger in the pure 2D setting. Since for small quench amplitudes the motion across the transverse $z$-direction remains frozen, $|E(\delta a_f)/N - E(\delta a_i)/N| \ll \epsilon_0$, these discrepancies have mainly two origins: deviations in the initial state and the distinct LHY correction terms.

The parametric dependence of the droplet's breathing frequency on $\delta a_f$ is displayed in Fig. 6(b) for two different transverse box sizes. Here, $\omega_b$ is measured from the spectrum of $\sigma_r(t)/\sigma_r(0)$ and it can be seen that, independently of the eGPE model and the value of $L_z$, the breathing frequency scales inversely proportional to $\delta a_f$, which is in accordance with the variational estimations of $\omega_b$ reported for 2D droplets [68]. Additionally, the disparities in $\omega_b$ among the two approaches are more prominent for smaller magnitudes of $\delta a_f$, since in this interaction regime the radial widths of the ground state droplets in 2D and quasi-2D have larger differences compared to $\delta a > 0$ (see Fig. 3(b)). This is also consistent with the smaller binding energies of the droplets in quasi-2D as compared to the 2D case, see Fig. 3(a).

For larger positive quench amplitudes, a significant initial droplet expansion occurs (Fig. 6(c)) within both eGPE models. At somewhat long evolution times e.g. around $1s$ in the case of $\delta a_f = 1.33 a_0$ in the quasi-2D setting, however, the droplet contracts again, thus signalling the onset of a breathing motion. This tendency for the time of the first contraction to become large for increasing $\delta a_f$ makes it experimentally difficult to observe the associated breathing dynamics, as instabilities such as three-body losses would set in. It is also worth noting that even for such large quench amplitudes, transverse excitations are highly suppressed since the pre- and postquench energy difference is still smaller than $\epsilon_0$. The observed differences are also caused by the distinct LHY contributions and deviations in

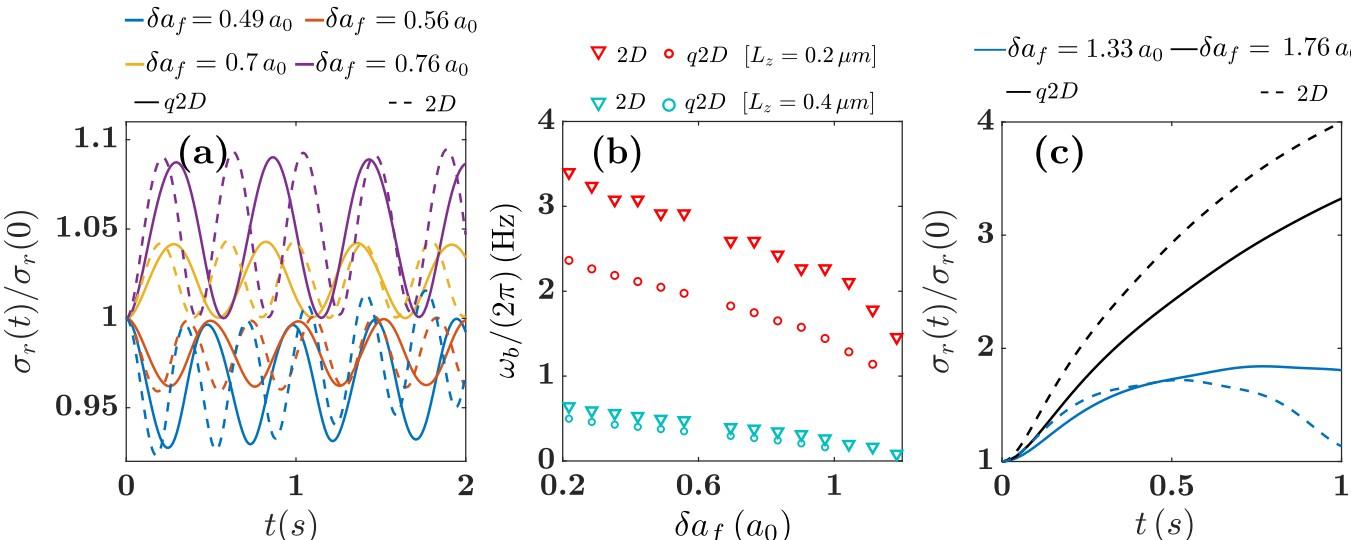

FIG. 6. (a) Time-evolution of the normalized radial width $\sigma_r(t)/\sigma_r(0)$ of the droplet for different post-quench interaction strengths $\delta a_f$. The oscillatory response demonstrates the contraction and expansion dynamics of the droplet. The dashed (solid) lines correspond to the prediction of the 2D (quasi-2D) eGPE model. (b) Dependence of the breathing frequency, $\omega_b$, on the post-quench mean-field interaction $\delta a_f$ for different transverse box sizes. Within the quasi-2D setting $\omega_b$ appears to be somewhat larger as compared to the 2D case. (c) Quenches to larger interactions $\delta a_f$ induce droplet expansion as can be also deduced from the growing tendency of $\sigma_r(t)/\sigma_r(0)$. The droplets contain $N = 2 \times 10^5$ atoms initialized in their ground state with $\delta a_i = 0.62\,a_0$ and being trapped in a box of length $L_x = L_y = 600\,\mu m$ and $L_z = 0.2\,\mu m$.

the initial state which are present.

The large quench amplitude adds larger energies $E(\delta a_f)/N - E(\delta a_i)/N$ into the system and therefore more complicated excitations can be anticipated in addition to the breathing mode [37, 68]. For instance, for the case with $\delta a_f = 1.76\,a_0$ shown in Fig. 7(b1)-(b5), the droplet features an initial uniform expansion and subsequently, from about $t > 300\,ms$, fragments into multiple ring structures which keep continuously expanding[2]. In fact, when $|E(\delta a_f)/N - E(\delta a_i)/N|$ is comparable to the chemical potential of the stationary state at $\delta a_f$, the energy pumped into the droplet due to the quench is comparable to its binding energy, causing it to break apart suggesting the involvement of self-evaporation. As expected this dynamical generation of rings occurs also within the 2D eGPE model but at shorter timescales. More concretely, for $\delta a_f = 1.33 a_0$ rings appear around $t = 200\,ms$ and consecutively expand followed by a contraction towards the center (not shown for brevity), reflecting the observed width contraction in Fig. 6(c), and a subsequent expansion tendency.

## V. IMPACT OF TRAP ANISOTROPY

In all experiments realized so far with homonuclear droplets [19, 20], atoms are held by means of external harmonic traps. While the impact of confinement on the stationary properties of droplets is a topic of ongoing investigation [38, 39, 77, 78], the understanding of harmonically trapped droplets in the dimensional crossover is far from complete. To study the effect of harmonic confinement in the quasi-2D geometry we introduce an anisotropic trap along the $x$-$y$ plane, with a weak trap of frequency $\omega_x = 2\pi \times 0.01\,\text{Hz}$ in the $x$ direction. Such a trap is characterized by an oscillator length of $160\,\mu m$, which is larger than the droplet width in the 2D and the quasi-2D settings (see Fig. 3(b)) and which therefore ensures that the trap impact is minimized. We then assume that the trap across the $y$ direction is tunable and has a frequency $\omega_y = \kappa \omega_x$, which means that the so-called aspect ratio $\kappa$ is a measure for how anisotropic the external potential is.

To account for the external trap, the 2D energy functional in Eq. (1) and the one in quasi-2D in Eq. (9) are amended

---

[2] In the presence of three-body recombination accounted for by the additional imaginary term in the eGPE $-i\hbar K_3/[2(1 + \sqrt{a_{22}/a_{11}})^3]|\Psi|^4\Psi$, we still observe the formation of ring structures and overall expansion until 0.8 seconds, while later on the droplet shrinks towards the center due to massive particle loss. Here, $K_3$ is the three-body recombination rate for the $|F = 1, m_F = 0\rangle$ hyperfine level [19].

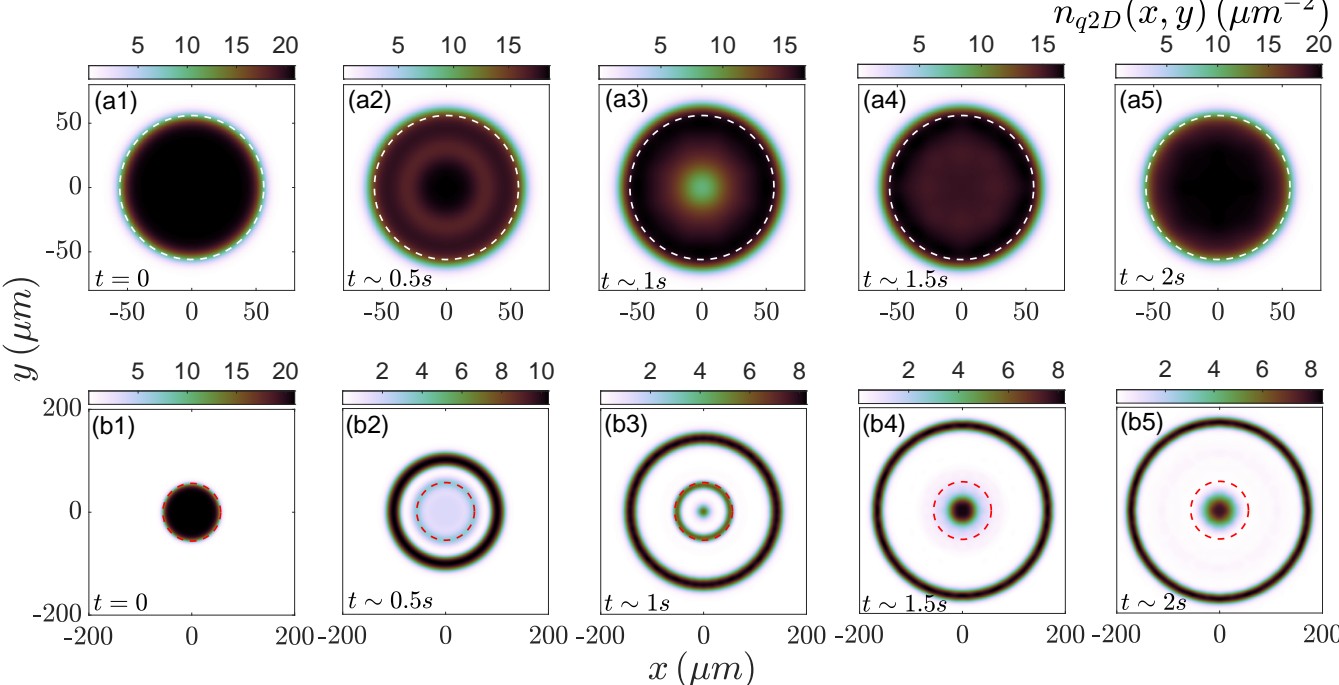

FIG. 7. Snapshots of the integrated (along the $z$-direction) density $n_{q2D}(x,y)$ following a quench of the averaged mean-field interaction from $\delta a_i = 0.62\,a_0$ to different $\delta a_f$. The droplet features (a1)-(a5) a breathing motion for $\delta a_f = 0.9\,a_0$ and (b1)-(b5) long-time expansion for $\delta a_f = 1.76\,a_0$ while fragmenting into multiple rings at later times. In all cases, the dashed circles designate the initial droplet edge. The remaining system parameters are the same as in Fig. 6.

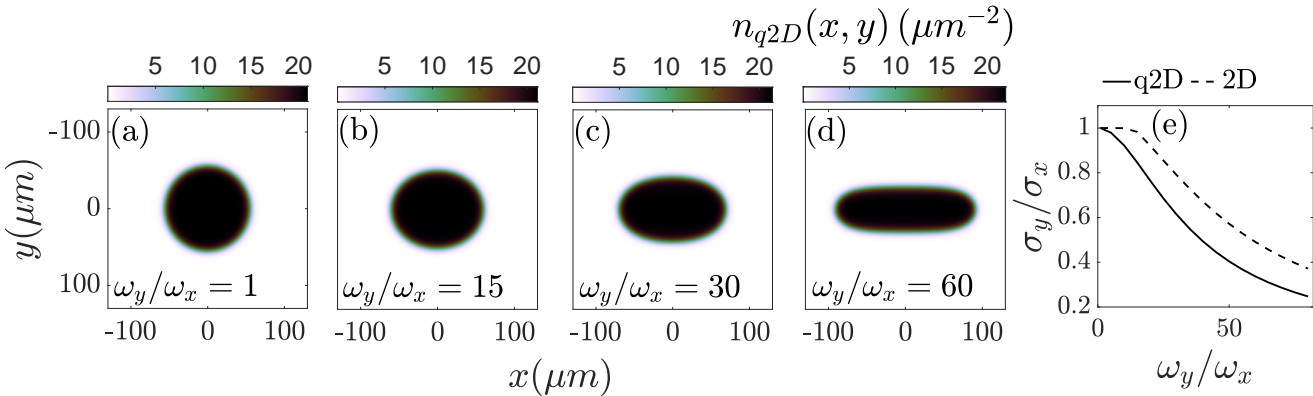

FIG. 8. (a)-(d) Ground state droplet density (integrated along the $z$ direction), $n_{q2D}(x,y)$, for various in-plane trap aspect ratios $\kappa = \omega_y/\omega_x$. For increasing $\kappa$, the quasi-2D droplet becomes squeezed (elongated) along the tightly (loosely) confined $y$ ($x$) direction while maintaining the same flat-top density value, manifesting its incompressible character. (e) Droplet width ratio, $\sigma_y/\sigma_x$, with respect to $\kappa$, showcasing that the impact of the trap in higher-dimensional droplets becomes substantial for larger aspect ratio. The droplet consists of $N = 2 \times 10^5$ bosons featuring an averaged mean-field interaction $\delta a_i = 0.62 a_0$. The trap frequencies correspond to $\omega_x = 2\pi \times 0.01$Hz and the box in the $z$-direction has a size of $L_z = 0.2\,\mu m$.

by the term

$$E_{\text{trap}}[\tilde{\Psi}] = \frac{m\omega_x^2}{2} \int d\tilde{\mathbf{r}}(x^2 + \kappa^2 y^2)\left|\tilde{\Psi}\right|^2. \tag{18}$$

Here, $\tilde{\Psi} \equiv \Psi(x,y,z)$ [$\tilde{\Psi} \equiv \psi(x,y)$] refers to the quasi-2D [2D] wave function, while $d\tilde{\mathbf{r}} \equiv d\mathbf{r}dz$ ($d\tilde{\mathbf{r}} \equiv d\mathbf{r}$) in the quasi-2D (2D) case. Furthermore, in the quasi-2D setup, the mixture remains transversely trapped by a box of length $L_z$, retaining the validity of the LHY correction given by Eq. (8). In the following, the trap effects will be investigated

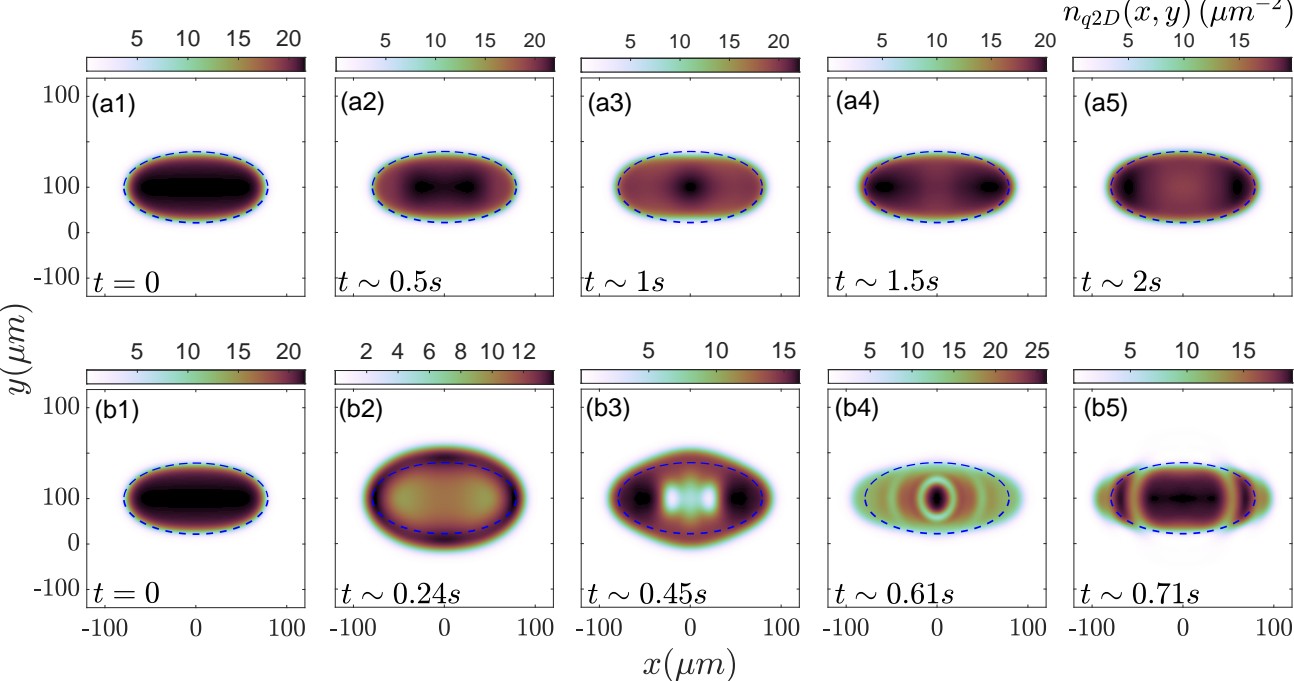

FIG. 9. Evolution of the integrated (along the $z$-axis) quasi-2D droplet density after an interaction quench from $\delta a_i = 0.62 a_0$ to (a1)-(a5) $\delta a_f = 0.9\,a_0$, and (b1)-(b5) $\delta a_f = 1.18\,a_0$. Dashed lines indicate the ground-state droplet edge for $\delta a_i = 0.62 a_0$ in the trap with anisotropy $\kappa = \omega_y/\omega_x = 40$. It can be deduced that a relatively small quench amplitude (panels (a1)-(a5)) excites both the quadrupole and the breathing modes, whereas large amplitude quenches (panels (b1)-(b5)) trigger the build-up of bulk and surface patterns. The remaining system parameters are the same as in Fig. 8.

for $\delta a_i = 0.62\,a_0$, and $L_z = 0.2\,\mu m$, where reasonably good agreement is found between the quasi-2D and 2D droplet models in the absence of a trap, see also Fig. 3(b).

For an isotropic trap, $\kappa = 1$, the integrated density profile $n_{q2D}(x, y)$ for the quasi-2D droplet is presented in Fig. 8(a) and a radial symmetry is visible similar to the free droplet illustrated in Fig. 7(a1). This is due to the relatively weak trap whose oscillator length in both directions is much larger than the droplet radial width in the absence of a trap. However, increasing the trap aspect ratio leads to the droplet becoming gradually compressed along the stronger confined $y$ direction, while being elongated along $x$ (see Fig. 8(b)-(d)). This spatial deformation of the droplet can be seen in the monotonic decrease of its width ratio $\sigma_y/\sigma_x$ as a function of $\kappa$ in Fig. 8(e), where $\sigma_i^2 = \frac{4}{N} \int d\mathbf{r} dz\, r_i^2 |\Psi(\mathbf{r}, z)|^2$ and $\mathbf{r} = (x, y)$. Importantly, despite the squeezing the constant flat-top density of the droplet remains mainly unaltered even for $\kappa = 80$, which can be seen from the constant color across the droplets in Figs. 8(a)-(d). This behavior provides another manifestation of the incompressibility of droplets [8].

The asymmetry in the density distributions appears to be less pronounced in the 2D model, especially for relatively smaller trap aspect ratios of $\kappa < 10$, as illustrated by the dashed line in Fig. 8(e). This can be understood by remembering that in the absence of external confinement, the radial droplet width in 2D is smaller compared to quasi-2D (see Fig. 3(b)). Therefore, the effect of a trap[3] can only be seen for larger frequencies and therefore in our case larger trap aspect ratios. As a consequence, $\sigma_y/\sigma_x$ in 2D remains always larger than in quasi-2D. We remark that differences between the two eGPE models occur also when inspecting the droplet binding energy (not shown for brevity). Particularly, in 2D the decrease rate of $|E|/N$ is slightly smaller ($\sim 4\,\%$) as $\kappa$ is adjusted from unity to 80, compared to the quasi-2D case ($\sim 6\,\%$).

The presence of the anisotropic trap also affects the droplets dynamical response to a quench. As an example we simulate an interaction quench for a quasi-2D droplet, going from $\delta a_i = 0.62\,a_0$ to $\delta a_f > \delta a_i$, in a trap with $\kappa = 40$. Similarly to the quench dynamics without a trap, the response of the system depends on the magnitude of the quench amplitude $\delta a_f - \delta a_i$.

---

[3] Naturally, a harmonic trap characterized by much larger frequencies leads to Gaussian-like droplets, and the energy per particle turns positive.

The time-evolution of the droplet density following a relatively small amplitude quench to $\delta a_f = 0.9\, a_0$ is depicted in Fig. 9(a1)-(a5). As can be seen, the planar droplet density, $n_{q2D}(x, y)$, squeezes and stretches in the course of the evolution along both directions. To understand the relative motion in the different directions, we note that $\sigma_y(t)$ and $\sigma_x(t)$ oscillate in an out-of-phase fashion. This indicates the excitation of a quadrupole mode, which can be favored by an anisotropic trap geometry. To confirm this we evaluate the width anisotropy $\langle \frac{x^2 - y^2}{x^2 + y^2} \rangle$ (t) of the planar density $n_{q2D}(x, y)$ [39], whose spectrum features a multitude of frequencies, with the most prominent quadrupole peak at $\omega_{qm}/(2\pi) = 0.28$ Hz. Notice that in the pure 2D geometry a similar quadrupole mode appears, but at a frequency $\omega_{qm}/(2\pi) = 0.39$ Hz, which is larger than in the quasi-2D case.

Besides the quadrupole mode, the quench also excites a small breathing motion in the density, which can couple to the quadrupole mode because of the broken symmetry [79]. To identify the breathing mode frequency for the example shown in Fig. 9, we calculate the spectrum of the droplet's radial width $\sigma_r$, and find $\omega_b/(2\pi) \simeq 2.05$ Hz. This breathing frequency is larger than the one in the absence of a trap, which is given by $\omega_b/(2\pi) \simeq 1.52$ Hz (see open circle in Fig. 6(b)). This discrepancy is due to the trap anisotropy, since for the isotropic case ($\kappa = 1$) the breathing frequency approaches the one of the untrapped droplet i.e. $\omega_b/(2\pi) = 1.56$ Hz. Recall here that we consider a very weak trap in the $x$ direction which essentially does not alter the droplet characteristics. On the other hand, the breathing frequency predicted by the 2D eGPE model is $\omega_b/(2\pi) = 2.49$ Hz, which is larger than the one in quasi-2D. This is again larger than the corresponding value in the absence of a trap, which is $\omega_b/(2\pi) = 2.26$ Hz (see open triangles in Fig. 6(b)). Finally, the differences between the breathing frequencies extracted from the 2D and the quasi-2D models can be attributed to the differences in the pre-quench ground states and different LHY contributions, especially since transverse excitations are absent due to the box energy $\epsilon_0$ being significantly larger than $|E(\delta a_i)/N - E(\delta a_f)/N|$.

Turning to quenches with comparatively larger amplitudes, we show an example with $\delta a_f = 1.18\, a_0$ in Fig. 9(b1)-(b5). One can immediately note an arguably larger amount of induced excitations compared to the previous case, which is a direct consequence of the increasing energy difference between the pre and postquench ground states. Indeed, the breathing and quadrupole modes feature more prominent amplitudes, as can be inferred from the enhanced structural deformations of the droplet compared to its initial (ground state) shape. In particular, the associated frequencies are found to be smaller for increasing quench amplitude, namely here they are $\omega_b/(2\pi) = 1.76$ Hz and $\omega_{qm}/(2\pi) = 0.24$ Hz respectively. Interestingly, apart from these collective modes prominent bulk excitations can be seen on top of the droplet density background. For example, a rhombic shape can be seen in Fig. 9(b3), while two-lobed, circular and parallelogram configurations appear in the bulk in Fig. 9(b2), (b4), (b5) respectively. Notice that despite the large quench amplitude, the density rings that form at later times (see Fig. 9(b4)) remain confined by the trap in contrast to the free space scenario. Let us also mention that the above bulk and surface excitations appear in the 2D model as well and are even found to be more pronounced (not shown). The appearance and consequent characterization of these surface and bulk excitations in the droplet is an intriguing prospect for future studies but lies beyond the scope of the present work.

## VI. SUMMARY AND PERSPECTIVES

In this work we have thoroughly investigated the ground state properties and dynamical response after an interaction quench of symmetric bosonic droplets in the crossover from two to three dimensions. This regime has been realised by considering a tight box confinement perpendicular to the plane and relying on experimentally realistic parameters. Our analysis is based on the direct comparison of various observables including the energy per particle, the droplet width and the particle density as predicted by the appropriate eGPEs in the 2D and the quasi-2D geometries. Additionally, we have constructed relevant variational approaches to shed further light on the droplet characteristics. Tuning the transverse box length has allowed us to consider distinct dimensionalities where the underlying LHY corrections are naturally modified. For example, in quasi-2D they contain a logarithmic coupling and also an additional density squared term. Examining the effects of the interplay of the involved mean-field interaction terms and the different LHY contributions as a function of the transverse box size and the mean-field coupling has therefore allowed us to first infer the parametric regions where the 2D description is accurate. In particular we have found that the logarithmic correction prevails for positive averaged mean-field interactions and a wide range of transverse box sizes, allowing one to regard droplets as purely two-dimensional. In this region the droplets exhibit relatively small binding energies and are significantly extended. On the other hand, for negative averaged mean-field interactions, the pure 2D description holds only for tight transverse confinements.

Quenching the averaged mean-field interaction by instantaneously changing the respective intra- and intercomponent couplings, we have shown that the droplet response predominantly consists of two distinct excitations. For relatively small quench amplitudes a collective breathing motion of the droplet is excited, together with small amplitude bulk excitations. The measured breathing frequency decreases for increasing post-quench interaction. In contrast, for large positive quench amplitudes the droplet shows long time expansion featuring pronounced ring shape excitation

patterns which eventually lead to the breaking of the self-bound state. In both cases the dynamics in the transverse direction remains practically frozen, due to the large box energy as compared to the excitation energy provided by the quench. Hence, deviations in the dynamical behavior of the droplets captured by the 2D and quasi-2D descriptions, e.g. disparities in the breathing frequency and timescales, can be attributed to differences in the initial state and inherently different LHY contributions among the two approaches.

Finally, we have explored the impact of an anisotropic trap on the ground state and the dynamics of a droplet. As expected, droplets feature prominent shape deformations due to the in-plane trap aspect ratio, with this behavior being generally more pronounced in the quasi-2D scenario, as long as the mean-field averaged repulsion is not strong. After an interaction quench, the presence of an anisotropic trap favors the formation of surface and bulk modes for increasing quench amplitude, while for relatively small ones the droplet undergoes a complex collective motion, having quadrupole and monopole components.

There are various questions that our work leaves open for future investigations. A straightforward one is to extend the present analysis to genuine two-component attractive mixtures where the emergent droplet phases are expected to be far richer [60, 62]. Another important direction is to examine the droplet crossover in one-dimension by employing the relevant eGPEs in realistic regimes, especially since corresponding experimental efforts are still lacking. Moreover, the inclusion of trap effects, being customary in experiments, and their impact on the LHY term as well as on the droplet formation is of significant interest. Certainly, it would be intriguing to further investigate the dynamical excitations of droplets in the presence of the trap asymmetry in order to controllably design pattern formation, e.g. similar to the ones triggered in repulsive gases [80, 81].

## ACKNOWLEDGEMENTS

This work was supported by the Okinawa Institute of Science and Technology Graduate University. The authors are grateful for the Scientific Computing and Data Analysis (SCDA) section of the Research Support Division at OIST. T.F. acknowledges support from JSPS KAKENHI Grant No. JP23K03290. T.F. and T.B. are also supported by JST Grant No. JPMJPF2221 and JSPS Bilateral Program No. JPJSBP120227414. S. I. M and G. B acknowledge fruitful discussions with P. G Kevrekidis, G. C. Katsimiga and I. A. Englezos on the topic of droplets.

## Appendix A: Comparison with the full 3D setting

In the main text we have discussed the parameter regions where the quasi-2D and 2D eGPE models are in agreement regarding the droplet characteristics. For completeness, we present in this Appendix a comparison of the quasi-2D and 2D models with the full 3D setting using a variational approach based on the respective energy functional with the appropriate LHY term. For this we first discuss the 3D energy functional and use a super-Gaussian variational ansatz to calculate the energies per particle in the droplet. These are then compared with the corresponding ones calculated in the quasi-2D and 2D approaches.

Similarly to the quasi-2D setting described in the main text, the original two-component bosonic system can be reduced to an effective single-component one by employing the condition $n_1/n_2 = \sqrt{a_{22}/a_{11}}$. In this case, the LHY energy density takes the form [7]

$$\mathcal{E}_{LHY}^{(3D)} = \frac{256\sqrt{\pi}}{15}\frac{\hbar^2}{m}|\Psi|^5 (a_{22}a_{11})^{5/4} f\left(\frac{a_{12}^2}{a_{11}a_{22}}, \sqrt{\frac{a_{22}}{a_{11}}}\right), \tag{A.1}$$

$$f(x,y) = \sum_{\pm}\left(1 + y \pm \sqrt{(1-y)^2 + 4xy}\right)^{5/2}\frac{1}{4\sqrt{2}(1+y)^{5/2}}, \tag{A.2}$$

where in the limit of $\delta a \to 0$ it holds that $f(1,y) = 1$, while for $\delta a < 0$, $f(x,y)$ is a complex function. However, it is possible to set $f(x,y) \simeq f(1,y)$ also for $\delta a < 0$, see Refs. [7, 11, 65]. The total energy functional is then given by

$$
\begin{aligned}
E_{3D}[\Psi] = \int dz d\mathbf{r} &\left[\frac{\hbar^2}{2m}|\nabla\Psi|^2 + \frac{\sqrt{a_{22}/a_{11}}}{(1+\sqrt{a_{22}/a_{11}})^2}\frac{4\pi\hbar^2\delta a}{m}|\Psi|^4 \right.\\
&\left. + \frac{256\sqrt{\pi}}{15}\frac{\hbar^2}{m}|\Psi|^5 (a_{22}a_{11})^{5/4} f\left(\frac{a_{12}^2}{a_{11}a_{22}}, \sqrt{\frac{a_{22}}{a_{11}}}\right)\right].
\end{aligned}
\tag{A.3}
$$

One can note that the second term stemming from the combination of the involved mean-field energies is identical to the equivalent contribution in the quasi-2D energy functional, see Eq. (9). Moreover, the third term of Eq. (A.3)

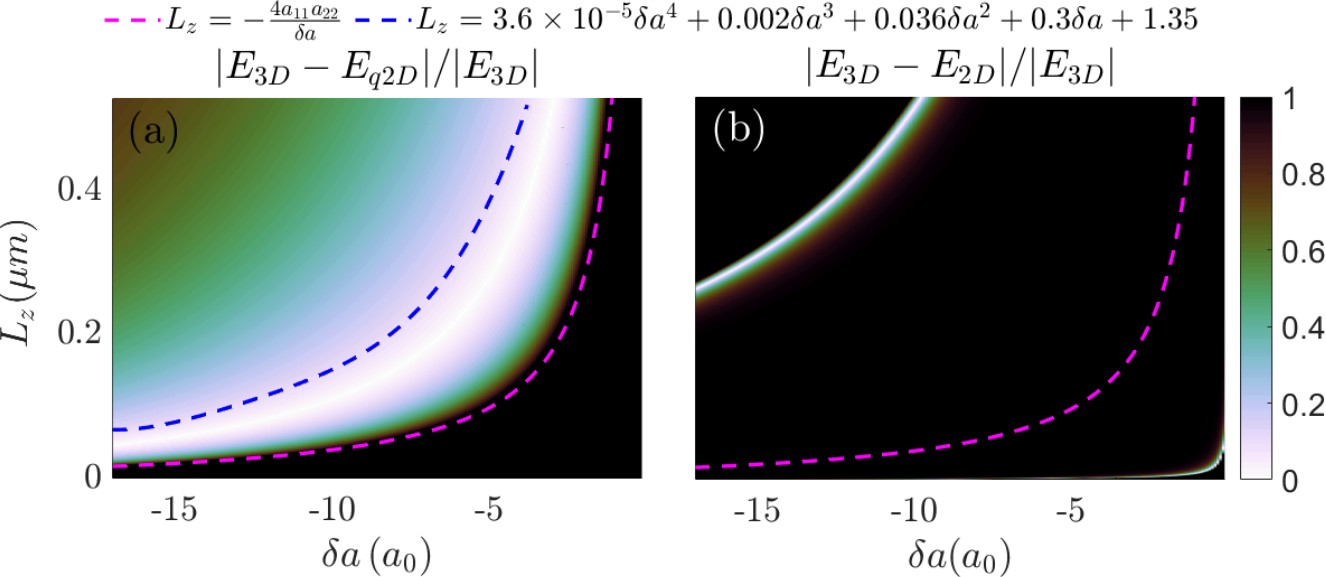

FIG. 10. Relative energy difference between the 3D eGPE variational model and (a) the quasi-2D and (b) the 2D ones as a function of the averaged mean-field interaction and the transverse box length. Good agreement between the 3D and the quasi-2D approaches is observed in the region between the magenta and blue dashed lines (see main text for their definition). Below (above) the magenta (blue) dashed line the 2D (3D) description is anticipated to be valid. On the other hand, no agreement between the 3D and 2D models can be seen (see text). Note that the colormap is bound at 1 in both figures as the relative difference can become large, thereby rendering the areas where agreement exists hard to discern.

refers to the 3D LHY correction. Despite the apparent differences with the quasi-2D LHY term (see Eq. (8)), it has been shown that in the thermodynamic limit, the two LHY energy contributions behave similarly for $\chi_{q2D} \sim 0.3$. Note, however, that the agreement between the LHY energy contributions extends to $\chi_{q2D} \gtrsim 0.3$ if one uses the aforementioned integro-differential equation [32].

We again assume that due to the tight transverse confinement in the crossover region, the 3D wave function can be approximated as a product of an in-plane and a perpendicular component, i.e. $\Psi(x, y, z) = \psi(x, y)\varphi(z)$, with $\varphi(z) = L_z^{-1/2}$. To explore the ground state energy dependence on the parameters $\delta a$ (note that in 3D droplets only exist for $\delta a < 0$) and $L_z$, we use a super-Gaussian ansatz for $\psi(x, y)$, which results in an energy per particle as

$$
\begin{aligned}
\frac{E_{3D}}{N} =& \frac{\hbar^2}{2m} \frac{m_r^2}{\sigma_r^2 \Gamma(1/m_r)} + \frac{N2^{-1/m_r}}{\pi \sigma_r^2 \Gamma(1 + 1/m_r)} \frac{\sqrt{a_{22}/a_{11}}}{(1 + \sqrt{a_{22}/a_{11}})^2} \frac{4\pi\hbar^2 \delta a}{mL_z} \\
&+ N^{3/2} \frac{256\sqrt{\pi}}{15L_z^{3/2}} \frac{\hbar^2}{m} (a_{22}a_{11})^{5/4} f\left(\frac{a_{12}^2}{a_{11}a_{22}}, \sqrt{\frac{a_{11}}{a_{22}}}\right) \frac{(2/5)^{1/m_r}}{\pi^{3/2}\sigma_r^3 \Gamma(1 + 1/m_r)^{3/2}}.
\end{aligned} \tag{A.4}
$$

The main difference between Eq. (A.4) and $E_{q2D}/N$ in a quasi-2D setting (see Eq. (14)) is the form of the LHY correction. Explicitly, it scales as $N^{3/2}\sigma_r^{-3}$ in 3D whereas in quasi-2D the logarithmic and high-density terms yield a $N\sigma_r^{-2} \log(\sigma_r^{-2})$ and a $N^2 \sigma_r^{-4}$ dependence respectively. Below, it will be argued that these terms are mainly responsible for the deviations in the droplet properties observed within the distinct models.

We compare the energy per particle in the different settings by showing the relative differences in Fig. 10(a). To quantitatively determine the boundary for the quasi-2D description to be valid we demand that the relative density deviations in the thermodynamic limit between the 3D and quasi-2D models are below 20%, which leads to the dashed blue line in Fig. 10(a). Using the same criterion for the quasi-2D and 2D models leads to the dashed magenta line. As it can be seen, the differences between $E_{3D}/N$ and $E_{q2D}/N$ are minimized in the region encased by the blue and magenta dashed lines, where the relative difference can be as low as $|E_{3D} - E_{q2D}|/|E_{3D}| \simeq 10^{-2}\%$ for $\delta a \approx -6.5\,a_0$ and $L_z \approx 0.2\,\mu m$. Indeed, this parameter region corresponds to $\chi_{q2D} \sim 0.3$, where the 3D and quasi-2D LHY terms are in good agreement [32]. However, the droplet energy per particle as predicted within the 3D and the quasi-2D models display large deviations ($|E_{3D} - E_{q2D}|/|E_{3D}| > 100\%$) below the magenta dashed line. On the other hand, above the blue-dashed line where the 3D LHY description is expected to be valid the deviations from the quasi-2D

model can be as high as $\sim 75\%$, e.g. for $\delta a \approx -16.5\,a_0$ and $L_z \approx 0.5\,\mu m$. These deviations in the energy per particle are not surprising, since below the magenta curve the bosonic mixture starts to behave as a purely 2D, while above the blue curve the quasi-2D LHY expression that is used here is only approximate and thus cannot correctly capture this parameter regime.

In contrast, poor agreement occurs between the 3D and 2D droplet energies across the whole parameter space as shown in Fig. 10(b). This can be expected and originates from the significant differences of the respective LHY terms in the underlying models. Therefore, the narrow bands appearing above and below the magenta dashed-line where the energy difference is small can be understood as purely accidental coincidences. There is no underlying physical explanation, since in the region above (below) the magenta-dashed line, the 2D (3D) description does not hold. This result further justifies the importance of the quasi-2D eGPE in capturing the droplet crossover from 3D to 2D.

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
