# Peer review of "Phases and dynamics of quantum droplets in the crossover to two-dimensions"

_SciPost Physics_

## Round 2 · Referee Report · Anonymous (Referee 1) · 2025-1-15

Report
Overall, the comments are adequately addressed, and I find that this has allowed for an improvement in the presentation. It is not entirely clear to me to what extent hard wall boundary conditions are relevant experimentally, as compared to the harmonic confinement, on the other hand, the justification provided by the authors is appropriate.
As a result, I recommend the publication of the article, while I also have a number of comments, which can optionally be taken into account.
Optional comments:
1) “One of our main findings is that droplets become substantially extended upon transitioning from negative-to-positive averaged mean-field interactions. This is accompanied by a significant reduction in their binding energies, which are approximately inversely proportional to the square of their size.” This is claimed to be the main finding, although there is not much discussion about it in the conclusions. 2) “We showcase” sounds to me like a ChatGPT phrase. 3) “We have thoroughly investigated” gives a positive opinion about the Authors' investigation, I would omit “thoroughly.”
Recommendation
Publish (easily meets expectations and criteria for this Journal; among top 50%)
Author: Jose Carlos Pelayo on 2025-03-17 [id 5294]
(in reply to Report 1 on 2025-01-15)
We thank the referee for their constructive remarks and for recommending our work for publication. As mentioned in our previous reply and also supported by relevant experimental references, box potentials can be readily implemented nowadays in cold atom experiments. Below, we address in detail each comment raised by the referee and provide a list of changes.
1) “One of our main findings is that ...
Our response: As suggested by the Referee, in the revised version of our manuscript, we have added a few relevant comments in the Summary and perspectives section, see also the list of changes.
2) “We showcase” sounds to me like a ChatGPT phrase.
Our response: We would like to emphasize that we did not use ChatGPT in generating any discussion in the manuscript. For this reason, and since we commonly use the word ”showcase”, we decided to keep it in our text.
3) “We have thoroughly investigated” gives a positive opinion about the Authors’ investigation, I would omit “thoroughly.”
Our response: We agree with the Referee and have accordingly omitted this word from the indicated place, see also the list of changes.
Author: Jose Carlos Pelayo on 2025-03-17 [id 5293]
(in reply to Report 2 on 2025-03-06)We once again thank the referee for their previous and current remarks that have improved the quality of our presentation. We are pleased to see that the referee now recommends the publication of our work in SciPost Physics.
We agree with the comment of the Referee and we have accordingly updated the units of the relative energy deviations into decimal form, see also the list of changes.

---

## Round 2 · List of Changes

• The abstract has been modified to emphasize that the droplet size increases as the averaged mean-field interactions transition from negative to positive values.
• In Sec. 2, in the first and second paragraphs, a discussion has been included regarding relevant experimental parameters and also explicating the fixed density ratio condition.
• On page 4, before Sec. 2.1, a sentence has been appended regarding the experimental feasibility of cylindrical box traps.
• On pages 4-5, Sec. 2.1 is now split into Sec 2.1.1 and Sec 2.1.2 providing a clear separation between the 2D and the quasi-2D geometries respectively.
• On page 5, Sec. 2.1.2, a brief discussion has been added to further clarify the definition of the χ-parameter.
• On Page 6, Eq. 8 was modified to have the correct units.
• On page 6 below Eq. 9, the discussion has been modified to clarify how E^{(1)}_{LHY} becomes negative as δa ≳ 0.
• On page 7, Eq. 12 has been modified as well as the equation in the footnote. Moreover, a discussion pertaining to n^{(q2D)}_0 has been added to clarify the quasi-2D density used in the rest of the manuscript.
• On page 7 just before Sec. 2.2, the discussion regarding the relative energy error is now amended.
• On page 7 after Eq. 13, the Gamma function Γ has been explicitly defined.
• On page 8, in the second paragraph after Sec. 3, the inequality symbol for the equilibrium density relative error has been changed from ≤ to ≲.
• On page 8, in the second paragraph of Sec. 3, a discussion has been included regarding the boundary below which the quasi-2D description becomes almost equivalent to the 2D one.
• The caption of Fig. 1 on page 9 has been modified to explicitly state that the shown curves refer to contour plots.
• On page 9, at the end of the first paragraph, the discussion about the dependence of the sign of the mean-field and LHY energy to the sign of δa now refers to the insets of Fig. 2 instead of just the main panels.
• On page 10, at the end of the paragraph, a discussion has been included regarding the nonmonotonous behavior of the energy per particle at negative averaged mean-field interactions.
• On page 10, we have updated the inset of Fig. 2(a) in order to clearly visualize that the individual energy contributions are different at L_z = 0.1μm.
• On page 13, the color scheme of Fig. 5 has been corrected to properly account for the various curves and be consistent with the legend.
• On page 13, in the middle of the first paragraph after Sec. 4, the definition of the transverse box-length, L_z , is provided again to ease reading.
• On page 14, in the caption of Fig. 6, a sentence has been appended regarding the comparison of the breathing frequency and the initial state’s chemical potential.
• On page 15, in the last paragraph, a discussion about the hard walls is added to emphasize that they are placed far enough such that the ring excitations do not reach them.
• On page 18, Fig. 9 has been updated to correct the y-tick labels.
• On page 21 in Appendix A, Fig. 10(b) has been revised, as well as the corresponding discussion in the final paragraph.
• All arXiv references in the manuscript were updated.

---

## Round 3 · List of Changes

• On page 7, we have modified the units of the relative energy deviation from percentages to decimals.
• On page 19, we deleted the word “thoroughly” at the beginning of the section Summary and perspectives.
• On page 19, a brief discussion has been added on the relation between the binding energy and the droplet size in the case of positive averaged mean-field interactions.
• On page 22, the value of the relative energy difference appearing in the inline equation is now in decimal form for consistency

---

## Editorial Decision

editorial_decision: